# Deriving and validating a risk prediction model for long COVID-19: protocol for an observational cohort study using linked Scottish data

Luke Daines [ID],[1] Rachel H Mulholland,[1] Eleftheria Vasileiou [ID],[1] Vicky Hammersley [ID],[1] David Weatherill,[1] Srinivasa Vittal Katikireddi [ID],[2] Steven Kerr,[1] Emily Moore,[3] Elisa Pesenti,[4] Jennifer K Quint [ID],[5] Syed Ahmar Shah [ID],[1] Ting Shi [ID],[1] Colin R Simpson,[1,6] Chris Robertson,[3,7] Aziz Sheikh[1]

LD and RHM contributed equally.

For numbered affiliations see end of article.

**Correspondence to**
Dr Luke Daines;
luke.daines@ed.ac.uk

## ABSTRACT

**Introduction** COVID-19 is commonly experienced as an acute illness, yet some people continue to have symptoms that persist for weeks, or months (commonly referred to as 'long-COVID'). It remains unclear which patients are at highest risk of developing long-COVID. In this protocol, we describe plans to develop a prediction model to identify individuals at risk of developing long-COVID.

**Methods and analysis** We will use the national Early Pandemic Evaluation and Enhanced Surveillance of COVID-19 (EAVE II) platform, a population-level linked dataset of routine electronic healthcare data from 5.4 million individuals in Scotland. We will identify potential indicators for long-COVID by identifying patterns in primary care data linked to information from out-of-hours general practitioner encounters, accident and emergency visits, hospital admissions, outpatient visits, medication prescribing/dispensing and mortality. We will investigate the potential indicators of long-COVID by performing a matched analysis between those with a positive reverse transcriptase PCR (RT-PCR) test for SARS-CoV-2 infection and two control groups: (1) individuals with at least one negative RT-PCR test and never tested positive; (2) the general population (everyone who did not test positive) of Scotland. Cluster analysis will then be used to determine the final definition of the outcome measure for long-COVID. We will then derive, internally and externally validate a prediction model to identify the epidemiological risk factors associated with long-COVID.

**Ethics and dissemination** The EAVE II study has obtained approvals from the Research Ethics Committee (reference: 12/SS/0201), and the Public Benefit and Privacy Panel for Health and Social Care (reference: 1920-0279). Study findings will be published in peer-reviewed journals and presented at conferences. Understanding the predictors for long-COVID and identifying the patient groups at greatest risk of persisting symptoms will inform future treatments and preventative strategies for long-COVID.

## INTRODUCTION

*In December 2019, an outbreak of a novel coronavirus was reported in Wuhan, China. The WHO declared the outbreak a global pandemic named*

---

### STRENGTHS AND LIMITATIONS OF THIS STUDY

⇒ We will use national data on ~99% of the Scottish population using the Early Pandemic Evaluation and Enhanced Surveillance of COVID-19 platform.

⇒ Our study will be unable to identify long-COVID patients who have not been in contact with healthcare services in Scotland.

⇒ Identifying long-COVID using routinely collected electronic health records may be challenging due to the lack of a standardised definition and variation in coding practices across healthcare systems.

⇒ To improve the identification of long-COVID (and associated clinical features), we intend to use free text in addition to the coded data available in electronic health records.

⇒ We are actively involving individuals who have experienced long-COVID to shape the research and ensure relevance to patients and the public.

---

*COVID-19 caused by the SARS-CoV-2 coronavirus. By November 2021, the WHO had reported over 240 million confirmed cases and at least five million deaths worldwide,[1] with more than nine million confirmed cases and over 140 000 deaths reported in the UK.[2]*

The severity and duration of the acute SARS-CoV-2 infection varies widely. Most people are asymptomatic or experience mild-to-moderate symptoms, while a smaller proportion (10%–15%) of cases experience more severe illness.[2] The majority of people recover after 2–6 weeks depending on disease severity.[3] However, some individuals have symptoms that last or recur for weeks or months after the initial acute infection.[3–18] Long-term effects of COVID-19 can present with a wide range of clinical features, relating to cardiovascular, neurological, respiratory and other organ systems, including mental

BMJ

health.[3–18] Common symptoms include fatigue, breathlessness, headaches, muscle weakness, joint pain and loss of taste or smell.[3 7 8 15–18]

Unified guidance to manage the long-term effects of COVID-19 in the UK has been developed by the National Institute for Health and Care Excellence (NICE), Scottish Intercollegiate Guidelines Network (SIGN) and the Royal College of General Practitioners (RCGP).[16] The guidance described two working case definitions of ongoing symptomatic COVID-19 (individuals with signs and symptoms of COVID-19 from 4 weeks to 12 weeks) and post-COVID-19 syndrome (individuals with signs and symptoms that develop during or following an infection consistent with COVID-19, continue for more than 12 weeks and are not explained by an alternative diagnosis).[16] The term 'long-COVID' therefore commonly refers to those who continue to present signs and symptoms 4 weeks after acute COVID-19 infection, that is, both ongoing symptomatic COVID-19 and post-COVID-19 syndrome.[16]

In Scotland, patients with symptoms suggestive of long-COVID are advised to seek medical care from their general practitioner (GP).[19] Diagnostic codes (Read codes, V.2) within the Scottish GP electronic system were introduced in March 2021 using NICE-led working definitions of long-COVID.[20] Equivalent diagnostic codes were also introduced in Scotland's Scottish Clinical Coding Standards using International Classification of Diseases 10th Revision (ICD-10) codes within secondary care data in February 2021.[21] The long-COVID diagnostic codes are available in online supplemental material.

Despite the progress in diagnostic coding, the prevalence and risk factors associated with long-COVID remain poorly understood, reflecting the lack of an agreed operational definition, the absence of diagnostic tests and the considerable variation in presentation. Reviews on the long-COVID literature have found that it was difficult to estimate the prevalence of persistent COVID-19 symptoms with certainty.[17 18] Therefore, alternative methods need to be adopted to identify those with long-COVID, so that the long-term consequences of COVID-19 illness can be better understood and individuals at highest risk of developing long-COVID can be identified early.[3 22–24] In this study, we aim to derive and validate a risk prediction model to estimate the probability that an individual will develop long-COVID. Our objectives are to: (1) create an operational definition of long-COVID through studying health system interactions using a national linked healthcare dataset; (2) derive and validate a risk prediction model to estimate the probability of developing long-COVID and (3) enhance the risk prediction model using machine learning.

## METHODS AND ANALYSIS
### Study design and population
We will undertake a national prospective population-based cohort study using the national Early Pandemic Evaluation and Enhanced Surveillance of COVID-19 (EAVE II) platform.[25 26] EAVE II comprises routinely collected primary care, secondary care, laboratory and serology data from 5.4 million Scottish residents registered with a GP (~99% of the Scottish population) from February 2020.[25 26] We will primarily focus on adults (aged ≥18 years) but will consider extending the cohort to include children (aged <18 years) if there are sufficient numbers of individuals in this age group. We intend to use data from February 2020 up to March 2023. The study started on 1 March 2021[20] and is scheduled to end on 28 February 2023.

### Inclusion/exclusion criteria
Since the baseline population for this study is everyone registered with a GP, those who are not registered with a GP in Scotland will be excluded from the analyses.

### Sample size calculation
We are using the whole population of Scotland and therefore sample size calculations are not applicable.

### Databases
The EAVE II platform links a wide range of routine healthcare datasets using pseudonymised identifiers of National Health Service (NHS) Scotland's Community Healthcare Index. We will use these routinely collected data sources (described below) to identify individuals with long-COVID and to determine their characteristics in the EAVE II cohort.

#### Primary care data
Primary care data will be extracted from GP practices via EAVE II's trusted third party Albasoft Ltd.[25 26] GPs in the UK provide healthcare services that are free at the point of service and usually act as the first point of contact into the healthcare system. This data source captures all clinical and administrative activity at GPs and the characteristics of registered patients. These data are stored either as: (1) clinical codes; or (2) written free text.[27] The latter is used to capture detailed information on any encounter and may provide additional information not available in coded data. In order to include data from primary care encounters when GP practices are closed, we will use out-of-hours (OOH) records derived from the Public Health Scotland (PHS) Primary Care OOH Data Mart.[26]

#### Secondary care data
Activity in hospital-based care will be extracted from the Scottish Morbidity Record (SMR) 01 which holds detailed information on hospital admissions, such as the specific area of clinical activity (specialty), the facility of care, patient management and new diagnoses.[28] Diagnoses in SMR01 will be extracted using ICD-10 codes.[28] For data on intensive care, we will use the Scottish Intensive Care Society Audit Group dataset of all adult patients admitted to intensive care units (ICUs) and high-dependency units in Scotland.[28] For outpatient care, we will use the SMR00

dataset, which captures outpatient activity in specialist clinics such as physiotherapy.[28]

## Laboratory data

All COVID-19 testing will be obtained from the Electronic Communication of Surveillance in Scotland dataset. This surveillance data contains all reverse transcriptase PCR (RT-PCR) tests, carried out in Scotland.[26] Sequencing data will be obtained from the Centre of Genomics and will make it possible to account for the variant of SARS-CoV-2 during model building.

## Vaccination data

COVID-19 vaccination data, including vaccination type and number of doses administered, will be available from two sources: GP records and the Turas Vaccination Management Tool, a web-based tool used to record community vaccinations in Scotland.[29]

## Telehealth data

Telehealth in Scotland is operated by NHS 24 Scotland, which delivers telephone and online services.[30] We are specifically interested in the NHS 24 111 teleservice, which provides OOH advice. During the pandemic, this service was expanded to include a COVID-19 helpline which was used to provide advice and triage patients to COVID-19 Assessment Centres.[30]

## Prescribing data

Prescription data relating to all medications prescribed and dispensed in the community in Scotland will be extracted from the prescribing information system.[26] These medications are coded using the British National Formulary (BNF) code lists.[31] For medication data within hospitals, Hospital Electronic Prescribing and Medicines Administration, which are available for five Health Boards will be used.[26]

## Mortality data

Mortality data will be taken from death registry data within the National Records of Scotland. These records hold information included on the death certificate, including cause(s) of death which are recoded using ICD-10 codes.[26]

## Other data

We will explore the use of other linkages available within the EAVE II platform. These include Scotland's Census 2011 from NHS Research Scotland for information on ethnicity, disability and occupation as part of the EAVE II substudy for ethnic and social inequalities in COVID-19 outcomes in Scotland.[32] We will also consider linkages and comparisons to Generation Scotland's CovidLife surveys which launched in April 2020 to capture how COVID-19 has been affecting volunteers in the UK.[33]

## Determining an operational definition for long-COVID

We will base our operational definition on the case definitions for the effects of COVID-19 illness at different time periods developed by NICE[16]:

1. Acute COVID-19 infection: individuals with signs and symptoms of COVID-19 for up to 4 weeks.
2. Ongoing symptomatic COVID-19: individuals with signs and symptoms of COVID-19 from 4 to 12 weeks.
3. Post-COVID-19 syndrome: individuals with signs and symptoms that develop during or following an infection consistent with COVID-19, continue for more than 12 weeks and are not explained by an alternative diagnosis. The post-COVID-19 syndrome usually presents with clusters of symptoms, often overlapping, which can fluctuate and change over time and can affect any organ system.

Long-COVID commonly refers to those who continue to present with signs and symptoms four or more weeks after acute COVID-19 infection, therefore, our primary outcome will include both ongoing symptomatic COVID-19 and post-COVID-19 syndrome. Our secondary outcome will focus on the clinical encounters suggestive of the post-COVID-19 syndrome. Further details are in the statistical analyses.

## Population characteristics

Population characteristics will be explored to assess the risk factors for developing long-COVID and to account for any confounding in our analyses.

### Sociodemographics

Age will be determined based on the available GP data and will be available as a continuous and categorical variable. Those aged over 100 will be truncated into the one group to overcome low sample size issues. Sex at birth will be included as a binary variable (female/male). Deprivation status will be derived from the Scottish Index of Multiple Deprivation 2020 quintile of the resident's postcode associated with their GP registration. Ethnicity data will also be included if completeness and quality of data is adequate. We will also consider other available information such as body mass index (BMI) and smoking status (smoker, ex-smoker, non-smoker and unknown).

### Geographical

Area of residence in terms of NHS Scotland Health Boards and local authorities will be considered. Settlement type will be determined by the urban/rural sixfold classification (UR6). Type of residence will also be considered such as private residence, care home and social/council housing if data are available.

### Clinical characteristics

Using diagnostic codes from the QCOVID algorithm,[34] we will identify the following conditions: (1) cardiovascular; (2) diabetes (type 1 and type 2); (3) respiratory; (4) cancer (blood cancer, chemotherapy, lung or oral cancer, marrow transplant, radiotherapy); (5) neurological; (6) other conditions, such as liver cirrhosis, osteoporotic fracture, rheumatoid arthritis, systemic lupus erythematosus, sickle cell disease, venous thromboembolism, solid organ transplant, renal failure (chronic kidney disease stages 3–5 with or without dialysis or transplant).[34]

### Severity of acute COVD-19 illness

Admission to hospital, any requirement for treatment in the ICU, and death will be used to categorise the severity of COVID-19 infection. We will define a COVID-19 hospitalisation as an RT-PCR confirmed positive test for SARS-CoV-2 in the 28 days prior to admission, or admission with an ICD-10 code for COVID-19. A COVID-19 ICU admission will be defined as a RT-PCR confirmed positive test for SARS-CoV-2 in the 28 days prior to ICU admission. A COVID-19 death will be defined as dying within 28 days of confirmed or probable COVID-19.

### Missing data

The amount of missing data will be examined for each variable of interest. Continuous variables, for example, BMI, will be imputed using predictive mean matching or imputation by chained equations if appropriate. Categorical variables with missing data will have a distinct group of 'Unknown'. We will consider dealing with these missing categorical variables by either keeping the distinct group, imputing them using chained equations or removing them from the analysis. The latter will be a complete-case analysis, which will reduce the total sample size.

### Statistical analyses

#### Developing an operational definition of long-COVID

We will first derive an operational definition for long-COVID by identifying patterns in clinical interactions within NHS Scotland services that may suggest long-COVID. A visual illustration of our intended methods is shown in figure 1.

#### *Indicators of long-COVID*

We are interested in (1) GP interactions; (2) hospital admissions; (3) outpatient attendances; (4) Accident and Emergency visits; (5) OOH encounters; (6) NHS 24 telehealth interactions; (7) medications (from GP prescribing and primary care pharmacy dispensing data) and (8) all-cause mortality. Our primary focus will be on the GP data, with other healthcare data providing corroborative information (step 1, figure 1). Information within these electronic health datasets will serve as an investigative list of potential indicators for long-COVID.

For the healthcare services (sources a to f), we will investigate the frequency of interactions and the reasons for each interaction which will include any new diagnoses (categorised by body system), treatments, tests or procedures related to long-COVID. For medications (7), we will investigate the frequency and type of new prescriptions using BNF chapters. For all-cause mortality (8), we will record the causes of death using ICD-10 codes. Figure 2 summarises the different data sources and potential indicators of long-COVID we intend to investigate. The dataset will comprise of categorical binary variables (eg, diagnosis or not) and numerical variables (eg, number of consultations).

GP records provide a rich source of primary care data, with the coded data providing additional context to interactions such as the type of interaction (eg, consultation, encounter, remote or face to face), referrals to specialty care and sick notes. For more detailed information on signs and symptoms that are indicative of long-COVID, we intend to use written free text available from GP records. We will use natural language processing (NLP) to identify key words or phrases of signs and symptoms relating to long-COVID. This NLP model will be applied to all written free text using Computer-Assisted (diagnostic) Coding. This will create derived codes associated with the key words and phrases (1 if the text mentions word

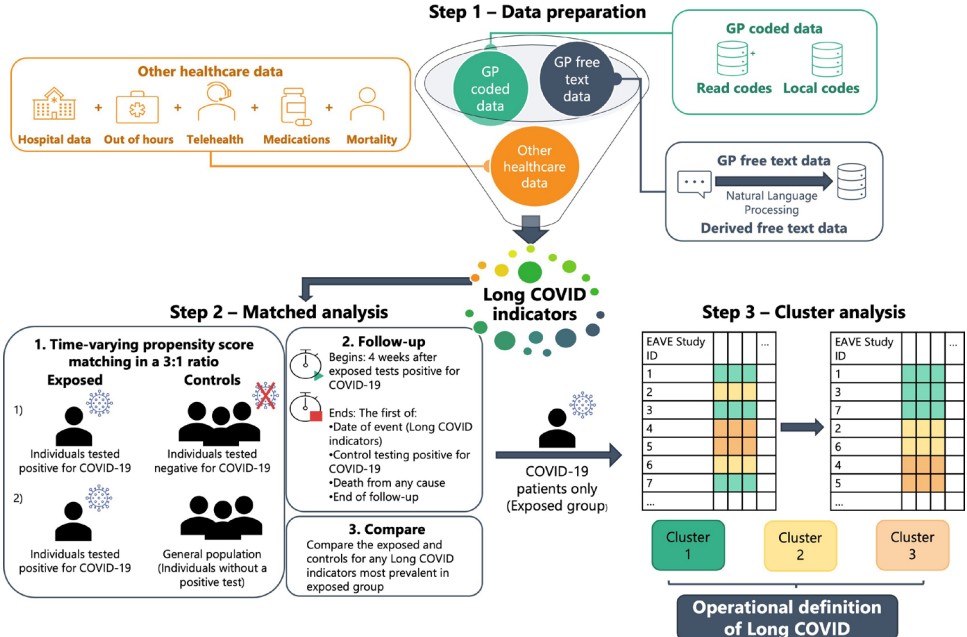

**Figure 1** Schematic diagram of methods for developing an operational definition for long COVID-19. EAVE, Early Pandemic Evaluation and Enhanced Surveillance of COVID-19.

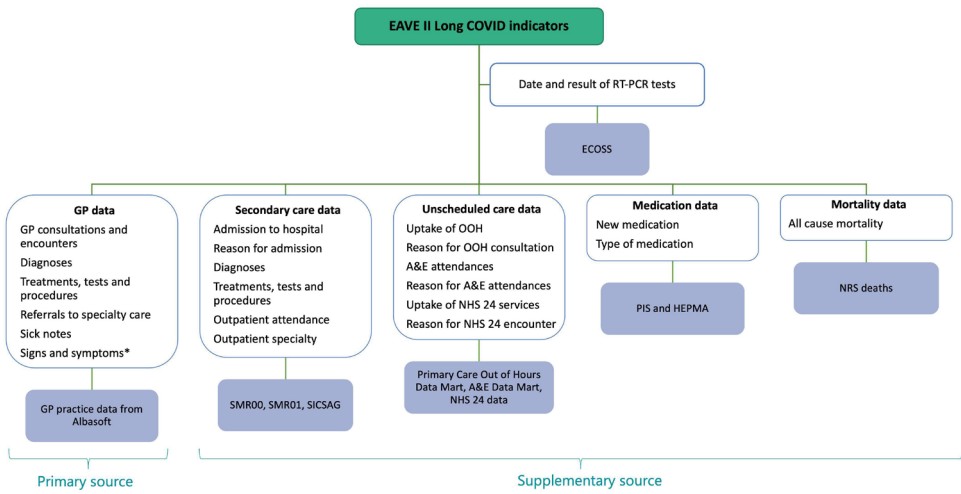

**Figure 2** Data linkage diagram of long-COVID indicators and their data sources within the EAVE II cohort. *Available through written GP free text. A&E, Accident and Emergency; ECOSS, Electronic Communication of Surveillance in Scotland; GP, general practitioner; HEPMA, Hospital Electronic Prescribing and Medicines Administration; NHS, National Health Service; NRS, National Records of Scotland; OOH, out-of-hours; PHOSP-COVID, The Post-hospitalisation COVID-19 study; PIS, Prescribing Information System; RT-PCR, real-time PCR; SICSAG, Scottish Intensive Care Society Audit Group; SMR, Scottish Morbidity Record.

or phrase, 0 otherwise).[35 36] These derived codes will be treated in a similar way to GP codes as discussed above. Figure 3 demonstrates this process of transforming the written GP free text into derived codes.

To initially explore potential long-COVID indicators, we will obtain summary level counts of all codes of interest within these healthcare datasets on individuals who tested positive and negative for COVID-19 using an RT-PCR test. We will count the frequency of these data ≥4 weeks after the date of the test. This will inform which codes relating to potential long-COVID indicators will be extracted on a patient level for further analysis.

### Matched analysis

To identify which of these long-COVID indicators are most important, we will perform a matched analysis (step 2, figure 1). The exposed group will be defined as the first date an individual tested positive for COVID-19 using an RT-PCR test. Two control groups will be assigned: (1) individuals who have had at least one negative RT-PCR test and have never tested positive up to the date of the exposed match testing positive and (2) the general population (everyone who did not test positive) of Scotland. These control groups will be investigated in turn.

We will use risk-set matching in a 3:1 ratio by time-varying propensity score matching. This will be based on the likelihood of testing positive for COVID-19 and will consider incorporating the following characteristics: sex, age, geography, comorbidities, risk factors, number of previous SARS-CoV-2 tests, deprivation status and urban–rural settlement. The adequacy of the matching will be assessed by checking for imbalance of the individual covariates across exposure groups.

For the matched analysis, each potential long-COVID indicator will be treated as its own dependent variable in turn. Follow-up will begin from 4 weeks after the exposed tested positive for COVID-19. Follow-up will end on either the date of event (if indicator is a binary variable), the control testing positive for COVID-19, death from any cause or the end of the follow-up period. Controls who have a positive test will be eligible to be included in the exposed group.

The long-COVID indicators will be compared between the exposed and control groups using statistical tests such as two-sample proportions test (for binary indicators), two-sample t-tests (for continuous indicators), Kaplan-Meier curves to inspect cumulative incidence, and survival analysis to look at the potential impact of interventions on long-COVID symptoms.

We also plan to conduct similar analyses on the whole cohort without propensity score matching. We will consider stratifying by age and sex if numbers allow.

### Cluster analysis

Clusters of long-COVID presentations in the exposed group will be investigated further, using the long-COVID

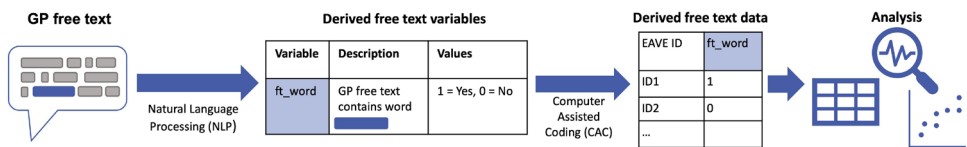

**Figure 3** Schematic diagram of NLP and CAC on GP free text. EAVE, Early Pandemic Evaluation and Enhanced Surveillance of COVID-19; GP, general practitioner.

indicators as our clustering input (step 3, figure 1). The indicators will be summarised using a window of four or more weeks after initially testing positive (eg, the number of interactions ≥4 weeks after the test). These indicators will not include the diagnostic codes for long-COVID since we are aiming to provide a more accurate alternative to this measurement of long-COVID.

We will explore both hierarchical clustering and k-means clustering, using distance measurements such as the Gower Distance, which is a suitable measurement of similarity for mixed categorical and numeric data.[37] We will also investigate clusters based on latent class analysis. We will then internally validate these clusters using statistics such as the silhouette coefficient and the Dunn index.[38 39] Comparisons between the clusters and long-COVID diagnostic codes will be undertaken for validation. The final set of clusters of long-COVID indicators will serve as our operational definition for long-COVID.

### Sensitivity analyses

We will perform a variety of sensitivity analyses to test the robustness of our long-COVID definition. This includes evaluating the start of follow-up to 12 weeks, to explore whether the alternative outcome definition of 'post-COVID-19 syndrome' display different clinical pathways. We will also investigate the patterns in the long-COVID indicators associated with the diagnostic long-COVID codes. To capture those who may be suffering from long-COVID but did not formally test positive for COVID-19 (or tested positive on a lateral flow device only), we will investigate the long-COVID indicators in the general population. We will also stratify by time period, for example, during the different peaks of positive cases in Scotland (eg, March 2020 to July 2020, August 2020 to April 2021, December 2021 to March 2022).[2] This will also reflect the dominant COVID-19 variants during the different waves of infection.

### Deriving and validating a risk prediction model for long-COVID

We will use the transparent reporting of a multivariable prediction model for individual prognosis or diagnosis (TRIPOD) guidelines to report the derivation and validation of the long-COVID prediction model (see completed checklist in online supplemental material).[40] The model will be derived using data from everyone in the cohort (defined above) who received a positive PCR test. We acknowledge the cohort may not include all people who had COVID-19 (for instance those who only tested positive by lateral flow device) but a positive PCR result is the most reliable marker of COVID-19 available from national datasets.

### Descriptive analysis

We will begin analysis by conducting descriptive analyses to visually inspect and summarise the types of potential long-COVID presentations within the clusters. Next, summaries of the geographical, sociodemographic and

risk factor profile of those presenting with the long-COVID clusters will be reported.

### Outcome

The outcome for the risk prediction model will be the derived operational definition of long-COVID defined from the cluster analysis. This will be dependent on the number of optimum clusters and the different classifications of long-COVID presentation. Depending on the clusters, we will classify our outcome into a binary variable of belonging to one (or more) cluster(s) (1) or otherwise (0).

### Predictor variables

Predictors for the risk prediction model will consist of the patient characteristics, including information on sociodemographics, geographical, clinical comorbidities and severity of COVID-19 infection. These will be a mixture of continuous, binary and categorical variables. Continuous variables will be tested for linearity and for more flexible relationships (using smooth splines). Groupings of continuous variables will be explored if necessary.

### Type of model

We intend to use a multivariable logistic regression model.

### Selection of predictors

We will build our model using stepwise selection based on the Akaike's information criterion and Bayesian information criterion. To assess the fit of the model parameters, the maximum likelihood ratio test will also be used.

### Model evaluation/performance

To evaluate the model's goodness of fit, we will use appropriate performance evaluation metrics such as the area under the Receiver Operating Characteristic (ROC) curve (captures the accuracy of the model discriminating between the outcome), and the calibration plot and slope (visualises the observed vs predicted values). Other evaluation measures such as the specificity (true negative rate), sensitivity (true positive rate) and accuracy will be considered if appropriate. We will also directly compare the predicted and observed values.

### Model validation

The model will be internally validated using k-fold cross validation. We will validate using different time periods as specified by one of the discussed sensitivity analyses in the clustering analysis. We will explore opportunities for external validation, comparison and meta-analysis with other long-COVID initiatives.

### Risk groups

To categorise the output of the model for further use by clinicians and COVID-19 patients, we will consider stratifying patients into risk groups based on the predictive probabilities in the multivariable model, for example, three groups of low, moderate and high risk of long-COVID.

### Sensitivity analysis

Sensitivity analyses for the risk prediction algorithm will depend on the outcomes from the sensitivity analyses from Objective 1 of developing an operational definition for long-COVID.

### Enhancing the prediction model using machine learning

Advances in machine learning will be utilised to enhance the development and validation of the prediction model. Specifically, we will systematically explore the use of supervised learning algorithms such as penalised models (eg, Least Absolute Shrinkage and Selection Operator (LASSO) regression), naïve Bayes classifier, gradient boosting decision trees and random forests to further improve the prediction model developed with traditional statistical methods. We will also consider using ensemble learning methods to strategically combine multiple models to obtain better predictive performance.

### Patient and public involvement

Lay input has shaped the development of this research and will continue throughout the project through the patient and public involvement (PPI) co-applicant, the EAVE II Public Advisory Group (PAG) and long-COVID Scotland. PPI members will collaborate with the research team to provide real-world perspectives when analysing and interpreting study findings ensuring that the work considers the needs, interests and concerns of patient and public members.

## ETHICS AND DISSEMINATION
### Ethical approval

This study forms part of the EAVE II project which is investigating epidemiological risk factors of COVID-19 disease. All data will be anonymised before being made available to the research team. EAVE II has already obtained permissions from the Research Ethics Committee (REC reference: 12/SS/0201), NHS Research and Development, GPs, NHS Health Boards and the Public Benefit and Privacy Panel (PBPP) for Health and Social Care (reference number: 1920-0279).

### Dissemination

To ensure the greatest impact of our findings, we will actively disseminate to three key audiences: policy/public health, academic and community based.

### Policy and public health

Findings from this project will provide evidence to help NHS Scotland and other international policy-makers identify groups of the population who are at most risk of long-COVID and related complications. We will work with partners in NHS Scotland, PHS and the Scottish Government to establish the best ways of disseminating these results and influencing policy. We also plan to disseminate findings through the National Core Studies programme, a UK government initiative supported by Health Data Research UK, in partnership with the Office for National Statistics.

### Academic

We will communicate our findings through presentations at major national and international scientific meetings and through publications in relevant peer-reviewed journals. We will publish our code and data dictionary on the EAVE II's GitHub repository (https://github.com/EAVE-II).

### Community based

We will write lay summaries of publications and create infographics to further communicate our findings via press releases, public and patient engagement events, social media and the EAVE II website (https://www.ed.ac.uk/usher/EAVE-ii).

**Author affiliations**
[1]Usher Institute, The University of Edinburgh, Edinburgh, UK
[2]MRC/CSO Social & Public Health Sciences Unit, University of Glasgow, Glasgow, UK
[3]Public Health Scotland, Glasgow and Edinburgh, UK
[4]Institute of Cell Biology, University of Edinburgh, Edinburgh, UK
[5]Faculty of Medicine, National Heart and Lung Institute, Imperial College London, London, UK
[6]School of Health, Wellington Faculty of Health, Victoria University of Wellington, Wellington, New Zealand
[7]Department of Mathematics and Statistics, University of Strathclyde, Glasgow, UK

**Acknowledgements** We are grateful for the support from Public Health Scotland (PHS) and Albasoft Ltd on the development of the protocol and data extraction. Methods were also developed with support from the CONVALESCENCE study analysts. We would also like to acknowledge the PPI support from the EAVE II PAG members and long-COVID Scotland.

**Contributors** AS conceived this manuscript. RM and LD led the writing of the manuscript with critical revision from EV, VH, DW, SVK, EM, EP, JKQ, TS and CRS. Statistical methods were reviewed by EAVE II's lead statistician CR and the other co-authors SVK, LD, SAS and SK. All authors reviewed the manuscript and gave final approval to be published.

**Funding** This work was supported by the Chief Scientist Office, grant number COV/LTE/20/15. EAVE II is supported by a grant (MC_PC_19075) from the Medical Research Council; a grant (MC_PC_19004) from BREATHE–The Health Data Research Hub for Respiratory Health, funded through the UK. Research and Innovation Industrial Strategy Challenge Fund and delivered through Health Data Research UK; a grant from the Data and Connectivity National Core Study, led by Health Data Research UK in partnership with the Office for National Statistics and funded by UK Research and Innovation (grant ref MC_PC_20058); Public Health Scotland; and the Scottish Government Director General for Health and Social Care. SVK acknowledges funding from a NRS Senior Clinical Fellowship (SCAF/15/02), the Medical Research Council (MC_UU_00022/2) and the Scottish Government Chief Scientist Office (SPHSU17).

**Competing interests** AS is a member of the Scottish Government Chief Medical Officer's COVID-19 Advisory Group and its Standing Committee on Pandemics. He is a member of the UK Government's Risk Stratification Subgroup and Astra-Zeneca's Thrombotic Thrombocytopenic Taskforce. All roles are unremunerated. SVK was co-chair of the Scottish Government's Expert Reference Group on Ethnicity and COVID-19 and a member of the UK Government's Scientific Advisory Group on Emergencies (SAGE) subgroup on ethnicity. All other authors declare no competing interests.

**Patient and public involvement** Patients and/or the public were involved in the design, or conduct, or reporting, or dissemination plans of this research. Refer to the Methods section for further details.

**Patient consent for publication** Not applicable.

**Provenance and peer review** Not commissioned; externally peer reviewed.

**ORCID iDs**
Luke Daines http://orcid.org/0000-0003-0564-4000
Eleftheria Vasileiou http://orcid.org/0000-0001-6850-7578
Vicky Hammersley http://orcid.org/0000-0002-9854-3868
Srinivasa Vittal Katikireddi http://orcid.org/0000-0001-6593-9092
Jennifer K Quint http://orcid.org/0000-0003-0149-4869
Syed Ahmar Shah http://orcid.org/0000-0001-5672-0443
Ting Shi http://orcid.org/0000-0002-4101-4535

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
