## [Reviewer comments · BMJ Open]

ARTICLE DETAILS

TITLE (PROVISIONAL)	Deriving and validating a risk prediction model for long COVID-19: protocol for an observational cohort study using linked Scottish data
AUTHORS	Daines, Luke; Mulholland, Rachel; Vasileiou, Eleftheria; Hammersley, Vicky; Weatherill, David; Katikireddi, Srinivasa; Kerr, Steven; Moore, Emily; Pesenti, Elisa; Quint, Jennifer; Shah, Syed Ahmar; Shi, Ting; Simpson, Colin; Robertson, Chris; Sheikh, Aziz;

VERSION 1 – REVIEW

REVIEWER	Löffler-Ragg, Judith Medical University of Innsbruck, Internal Medicine II
REVIEW RETURNED	06-Jan-2022

GENERAL COMMENTS	The management of long-term sequelae after COVID-19 has growing importance for health care systems. The prevalence and risk factors associated with long COVID are poorly understood. This excellent study protocol aims at deriving and validating a risk prediction model for long COVID-19 based on the entirety of Scottish health data. They will use the national Early Pandemic Evaluation and Enhanced Surveillance of COVID19 (EAVE II) platform, a population-level linked dataset of routine electronic healthcare data from 5.4 million individuals in Scotland. The results of epidemiological studies are very dependent on complete and standardized datasets, but corresponding limitations have been mentioned and are tried to be overcome by data mining from different approaches (GPs, hospital admissions, outpatient attendances, prescriptions, data on mortality.. etc. Appropriate control groups are also included in this prospective observational trial. The methodological approach is therefore very well thought out, and the statistics including machine learning seems to be a sensible approach for this large data set. Such a publication of the entirety of real-world data on Long COVID does not exist. This project has the potential to map complete epidemiological real-world data for a region, to filter out risk factors and to develop prediction models. All necessary details are provided, except the trial registration was not apparent to me. It was an honor to review this, from my point of view the protocol can be accepted for publication without any changes, however, because of its complexity it might still be useful to get a statistician's opinion.
--

REVIEWER	Sillesen, Martin Copenhagen University Hospital, Organ Surgery and Transplantation
REVIEW RETURNED	03-Mar-2022

GENERAL COMMENTS	Thank you for the opportunity to review this interesting protocol. There is no doubt that the area of study is important, and the authors should be commended for submitting a thorough protocol prior to study commencement. This will greatly assist in the transparency of the study to come. Overall, I found the protocol concise and well written. My comments are thus only of a minor character: 1) During the study period, multiple different SARS-CoV2 viral strains have been prevalent, although it is unclear whether this affects long-COVID risk. Will you be able to control for viral strain type in your model? It may not be possible, but it would benefit the protocol to clarify this. 2) Differences in types of vaccinations (e.g. Pfizer, Glaxo etc), as well as number of vaccinations given prior to the COVID diagnoses could potentially have an effect as well. Will you be able to account for this? In any case, it may be worthwhile updating the protocol with this information. 3) In Scotland, is a PCR mandatory for COVID diagnosis? In some countries, a quicktest is sufficient if the patient have mild symptoms (non hospitalised). Updating the protocol with this information could improve readability 4) A problem that could arise, is that fluctuations in factors not possible to account for in the models could affect results. As such, Long-COVID was not known in early 2020, and a number of patients are thus unlikely to have consulted a doctor for symptoms. Also, at different stages of the pandemic, the adherence to PCR testing may fluctuate, and the number of COVID patients without a confirmatory PCR diagnoses may differ widely, especially in light of the dynamic updates on testing requirements. This is of course almost impossible to account for in a retrospective protocol like this one, but two things could be considered:  - Could it be worthwhile to introduce subgroups from the different waves of the pandemic (e.g. winter 2020, winter 2020-2021 and winter 2021-2022). You may get a different signal from the different waves, which could provide interesting information and in part account for the fluctuations in testing strategies etc. - Many Long-COVID sufferers are likely to be either mis or undiagnosed. An interesting approach would be to screen the healthcare data for new onset symptoms/diagnoses that match those of long-covid in patients without a positive PCR. While one can of course not conclude that these diagnoses are COVID related, it could provide some interesting insight into the scale of the dark (non-diagnosed) numbers. This again could be worked into the predictive models as a supplementary. It could be that these supplementary models actually provided a better approximation for the scale of the problem. Again, I would like to congratulate the authors for an excellent protocol, I look forward to reading the study results once these are available.
--

VERSION 1 – AUTHOR RESPONSE

Reviewer 1 (Professor Judith Löffler-Ragg):

The management of long-term sequelae after COVID-19 has growing importance for health care systems. The prevalence and risk factors associated with long COVID are poorly understood. This excellent study protocol aims at deriving and validating a risk prediction model for long COVID-19 based on the entirety of Scottish health data. They will use the national Early Pandemic Evaluation and Enhanced Surveillance of COVID10 19 (EAVE II) platform, a population-level linked dataset of routine electronic healthcare data from 5.4 million individuals in Scotland.

The results of epidemiological studies are very dependent on complete and standardized datasets, but corresponding limitations have been mentioned and are tried to be overcome by data mining from different approaches (GPs, hospital admissions, outpatient attendances, prescriptions, data on mortality, etc. Appropriate control groups are also included in this prospective observational trial.

The methodological approach is therefore very well thought out, and the statistics including machine learning seems to be a sensible approach for this large data set.

Such a publication of the entirety of real-world data on Long COVID does not exist.

This project has the potential to map complete epidemiological real-world data for a region, to filter out risk factors and to develop prediction models. All necessary details are provided, except the trial registration was not apparent to me.

Thank you. We appreciate your suggestion for registering the study as a trial, but as the design is a retrospective matched cohort we do not feel that trial registration is necessary.

It was an honour to review this, from my point of view the protocol can be accepted for publication without any changes, however, because of its complexity it might still be useful to get a statistician's opinion.

Thank you very much.

Reviewer 2 (Dr Martin Sillesen):

Thank you for the opportunity to review this interesting protocol. There is no doubt that the area of study is important, and the authors should be

commended for submitting a thorough protocol prior to study commencement. This will greatly assist in the transparency of the study to come. Overall, I found the protocol concise and well written.

Thank you.

1) During the study period, multiple different SARS-CoV2 viral strains have been prevalent, although it is unclear whether this affects long-COVID risk. Will you be able to control for viral strain type in your model? It may not be possible, but it would benefit the protocol to clarify this.

EAVE II has access to sequencing data via the Centre of Genomics and therefore we hope to be able to take account for the variant of SARS-CoV-2 during the modelling. We have added the following text on page 6:

“Sequencing data will be obtained from the Centre of Genomics (COG) and will allow the variant of SARS-CoV-2 to be accounted for during model building.”

2) Differences in types of vaccinations (e.g. Pfizer, Glaxo etc), as well as number of vaccinations given prior to the COVID diagnoses could potentially have an effect as well. Will you be able to account for this? In any case, it may be worthwhile updating the protocol with this information.

Yes, we intend to use data on vaccination status, which includes the number of doses and type of vaccination given. We have updated the manuscript as follows (page 6):

COVID-19 vaccination data, including vaccination type and number of doses administered, will be available from two sources: GP records and the Turas Vaccination Management Tool (TVMT), a web-based tool used to record community vaccinations in Scotland.[29]

3) In Scotland, is a PCR mandatory for COVID diagnosis? In some countries, a quick test is sufficient if the patient have mild symptoms (non hospitalised). Updating the protocol with this information could improve readability

In Scotland, a PCR test is not mandatory for a COVID diagnosis. Guidance on how to confirm a COVID-19 diagnosis has changed during the pandemic and PCR has been the most reliable and consistently used test for individuals with symptoms or for those requiring medical assessment. We

acknowledge that relying on PCR to indicate COVID-19 will miss individuals who have tested positive for COVID-19 via a lateral flow device, or not been tested at all. However, as the results of lateral flow tests are not routinely recorded in a national dataset there is no reliable way of identifying individuals who have tested positive for COVID-19 by lateral flow test alone. In light of your comments we have added the following (page 12 and)

*“The model will be derived using data from everyone in the cohort (defined above) who received a positive PCR test. **We acknowledge the cohort may not include all people who had COVID-19 (for instance those who only tested positive by lateral flow device) but a positive PCR result is the most reliable marker of COVID-19 available from national datasets.**”*

And

*“To capture those who may be suffering from long-COVID but did not formally test positive for COVID-19 **(or tested positive on a lateral flow device only)**, we will investigate the long-COVID indicators in the general population.”*

4) A problem that could arise, is that fluctuations in factors not possible to account for in the models could affect results. As such, Long-COVID was not known in early 2020, and a number of patients are thus unlikely to have consulted a doctor for symptoms. Also, at different stages of the pandemic, the adherence to PCR testing may fluctuate, and the number of COVID patients without a confirmatory PCR diagnoses may differ widely, especially in light of the dynamic updates on testing requirements. This is of course almost impossible to account for in a retrospective protocol like this one, but two things could be considered:

Could it be worthwhile to introduce subgroups from the different waves of the pandemic (e.g. winter 2020, winter 2020-2021 and winter 2021-2022). You may get a different signal from the different waves, which could provide interesting information and in part account for the fluctuations in testing strategies etc.

Yes, we agree with you and did plan to conduct sensitivity analyses as you suggest. We have made the following change on page 12:

*We will also stratify by time-period, for example during the different peaks of positive cases in Scotland (e.g., March 2020 to July 2020, August 2020 to April 2021, **December 2021 to March 2022**).[4] This will also reflect the dominant COVID-19 variants during the different waves of infection.*

Many Long-COVID sufferers are likely to be either mis or undiagnosed. An interesting approach would be to screen the healthcare data for new onset symptoms/diagnoses that match those of long-covid in patients without a positive PCR. While one can of course not conclude that these diagnoses are

COVID related, it could provide some interesting insight into the scale of the dark (non-diagnosed) numbers. This again could be worked into the predictive models as a supplementary. It could be that these supplementary models actually provided a better approximation for the scale of the problem.

We agree that this could be an interesting analysis and had actually included it in our original draft. We have clarified our thoughts as follows (page 12)

“We will perform a variety of sensitivity analyses to test the robustness of our long-COVID definition. This includes evaluating the start of follow-up to 12 weeks, to explore whether the alternative outcome definition of ‘post-COVID-19 syndrome’ display different clinical pathways. We will also investigate the patterns in the long-COVID indicators associated with the diagnostic long-COVID codes. To capture those who may be suffering from long-COVID but did not formally test positive for COVID-19 (or tested positive on a lateral flow device only), we will investigate the long-COVID indicators in the general population.”

VERSION 2 – REVIEW

REVIEWER	Sillesen, Martin Copenhagen University Hospital, Organ Surgery and Transplantation
REVIEW RETURNED	11-Apr-2022
GENERAL COMMENTS	Thank you for responding to my points mentioned in the review. All questions have been fully answered. I look forward to reading the results of this interesting study.